# Experimental Study on Creep Characteristics and Long–Term Strength of Anthracite

**Jianbing Yan [1], Xiaoqiang Zhang [2,*], Kai Wang [2], Xuanmin Song [1], Shaofei Yue [1] and Jian Hou [2]**

[1] Key Laboratory of In-Situ Property Improving Mining of Ministry of Education, Taiyuan University of Technology, Taiyuan 030024, China
[2] College of Mining Engineering, Taiyuan University of Technology, Taiyuan 030024, China
* Correspondence: zhangxiaoqiang@tyut.edu.cn

**Abstract:** Coal pillars from old mines undergo creep, a type of time−dependent deformation. Research on underground engineering stability of coal mines has increasingly focused on long−term strength as an important mechanical index of the creep characteristics of coal pillars. This study performed conventional triaxial compression and triaxial creep tests of anthracite under different confining pressures. The creep law of anthracite and the long−term strength of anthracite was studied according to the test results. The test results demonstrated the following: (1) The conventional elastic modulus and peak strain of anthracite increased exponentially with the confining pressure. (2) Under low stress levels, anthracite exhibited only instantaneous deformation and attenuation creep. In contrast, anthracite exhibited instantaneous deformation, attenuation creep, and steady creep under high stress levels; accelerated creep occurred until failure when the stress reached a particular value. (3) By fitting the steady−state creep rate of anthracite under high stress levels, the functional relationship between axial stress and steady−state creep rate was established, and the threshold of the steady−state creep rate in high−stress−level areas was suggested as the optimum long−term strength of anthracite. (4) The ratio of the long−term strength to instantaneous strength under various confining pressure grades ranged from 70% to 91%.

**Keywords:** creep; anthracite; test; long−term strength; improved steady−creep rate method





## 1. Introduction

Before the 1990s, many mines in the Qinshui coalfield in China mostly adopted the roadway pillar (digging instead of mining) and roadway caving mining methods, leaving many coal pillars in the abandoned mining areas [1]. With the gradual depletion of coal resources, large−scale residual coal recovery from high−quality coal resources in old mining areas has become necessary; however, long−term stability discrimination of coal pillars remaining in old mining areas is a problem that must be considered for the safe recovery of residual coal [2–4]. Coal pillars have time−dependent creep characteristics. Coal pillars from old mines have been subjected to roof loads for a long time, and their creep characteristics are more apparent. The creep deformation process of coal pillars has a critical stress value. When the load on the coal pillar is greater than this critical value, the coal pillar stabilises with time and creep is accelerated until instability and failure. In contrast, when the load on the coal pillar is less than the critical value, the coal pillar remains intact regardless of time. In rock mechanics, this critical value is known as long−term strength [5–8]. From the initial formation to the recovery of coal pillars, whether the deformation of the coal pillar was within the safe allowable range with an increase in time or reached long−term strength, lost stability, and was destroyed should be considered when studying the creep of coal pillars. In addition, the stress environment of the elastic core area inside the coal pillar is generally a low−confining−pressure triaxial stress state. Therefore, triaxial creep characteristics under low confining pressure must be investigated,

and long−term strength tests must be performed to predict the long−term stability of coal pillars.

In recent years, many scholars have primarily studied the creep law and constitutive model of various rocks under different conditions. Zhang et al. [9] performed compression and creep tests on peridotite and concluded that creep deformation occurs only when the deviatoric stress is higher than the creep starting threshold. Chen et al. [10,11] performed triaxial creep tests on quartz, metamorphic, and red sandstones under freezing and thawing at different temperature differences and discussed their pore structure development law and creep damage characteristics. Zhang et al. [12] studied the triaxial creep test law of salt rock at different temperatures and established a fractional viscoelastic–plastic creep damage model of salt rock considering the temperature effect. Jiang et al. [13] performed creep tests on gypsum rock under different confining pressures and obtained its creep deformation characteristics under low− and high−stress conditions. Cheng et al. [14] performed uniaxial compression and graded creep tests on cemented backfill in coal mines by ultrasonic monitoring and established the hardening–damage nonlinear constitutive model of cemented backfill by introducing hardening and damage functions. Other researchers have also performed creep tests on soft rock [15], granite [16], clay rock [17], and siltstone [18] among others. They discussed the creep laws of various rocks under different conditions, such as creep strength and creep deformation changing with stress, and established creep models of various rocks according to the test data.

As an important mechanical index of rock mechanics, long−term strength has been the research focus in creep tests. The commonly used methods for determining the long−term strength of rock include the isochronous stress–strain curve, steady creep rate, and transition creep methods [19,20]. Many researchers have conducted studies on and made improvements to these methods. According to the uniaxial creep test results of red sandstone, Cui et al. [21] suggested that the critical stress of rock entering radial steady creep should be considered as the long−term strength of rock. Zhang et al. [22] improved the isochronous stress–strain curve method and proposed a method that considers the critical stress value of rock lateral volume expansion rate greater than the axial compression rate to be long−term strength. Liu et al. [23] separated the creep stages of rock, established the functional relationship between steady creep rate and stress, and took the stress value corresponding to the creep rate tending to zero as the long−term strength of rock. Wang et al. [24] improved the traditional steady−state creep rate method to determine the long−term strength of salt rock, taking the reciprocal of stress level and steady−state creep rate as independent variables and strain as function curves. They proposed that the stress corresponding to the linear fitting intersection point of high− and low−stress areas should be considered the long−term strength of rock.

Although many scholars have proposed improved methods for determining the long−term strength of various types of rocks, these methods were inconsistent, and the strength values were irregular owing to the discreteness of engineering rock mass and the uniqueness of its internal structure. Therefore, this study performed triaxial compression instantaneous and creep tests under different confining pressures on anthracite taken from coal seam No. 3 of the Qinshui coalfield in Shanxi Province and obtained the relationship between axial deformation law, creep deformation rate, and stress level. Moreover, the long−term strength determination method is discussed. The results of this study provide a relevant theoretical reference for the study of coal creep mechanical properties, long−term stability evaluation of coal pillars left behind in coal mines, and control of surrounding rock in re−mining stopes.

## 2. Sample Preparation and Test Equipment

### 2.1. Sample Preparation

The coal samples were taken from coal seam No. 3 in the re−mining area of the Guanlingshan Coal Mine in the Qinshui coalfield, as shown in Figure 1a. Coal samples with a block size of approximately 300 mm × 300 mm × 300 mm were drilled on site

using pneumatic picks, as shown in Figure 1b, The samples were wrapped tightly with plastic wrap to prevent contact with air, protected and packaged with sawdust and foam, and transported to the laboratory. The coal bodies were processed into standard cylinder samples of $\Phi$ 50 mm × 100 mm using a yarn−cutting machine according to the International Society of Rock Mechanics (ISRM) test standards, as shown in Figure 1c. The two ends of the coal samples were ground using an end grinder to ensure parallelism between the upper and lower ends, as shown in Figure 1d. Considering the discreteness of the anthracite samples, the sonic testing method was used to screen coal samples with similar wave velocities before triaxial compression and creep tests, as shown in Figure 1e. Figure 1 shows the sample processing and screening process.

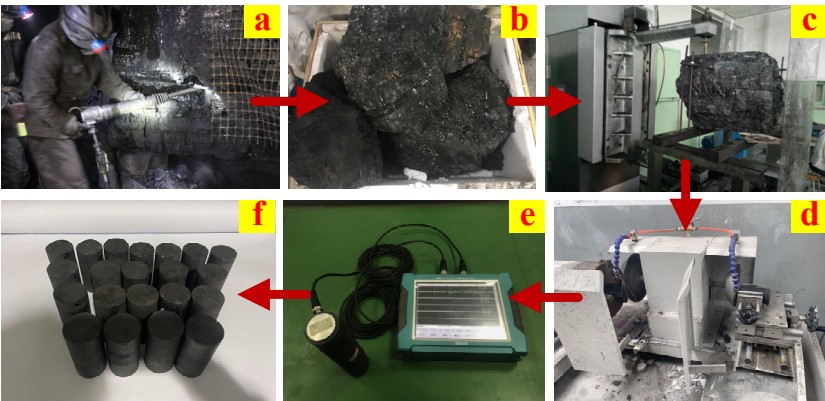

**Figure 1.** Coal sample preparation and specimens ready for testing.

### 2.2. Test Equipment

Triaxial compression instantaneous and creep tests were performed using an LDHJ−III rock triaxial high−temperature creep−testing machine jointly developed by the Taiyuan University of Technology and Jiangsu Huaan Scientific Research Instrument Chief, as shown in Figure 2. The maximum axial pressure of the device was 300 kN, and it had a measuring accuracy of less than ±0.1 kN. The maximum confining pressure was 20 MPa, with the accuracy controlled within ±0.02 MPa. The measuring deformation range was 0–10 mm in the axial and radial directions. All measuring resolutions were at 0.001 mm. The temperature range was 5–80 °C, with the temperature control accuracy at 0.2 °C. To provide creep parameters for studying the long−term stability of coal pillars, the test temperature was set according to the actual temperature of the coal pillar stratum of 20 °C.

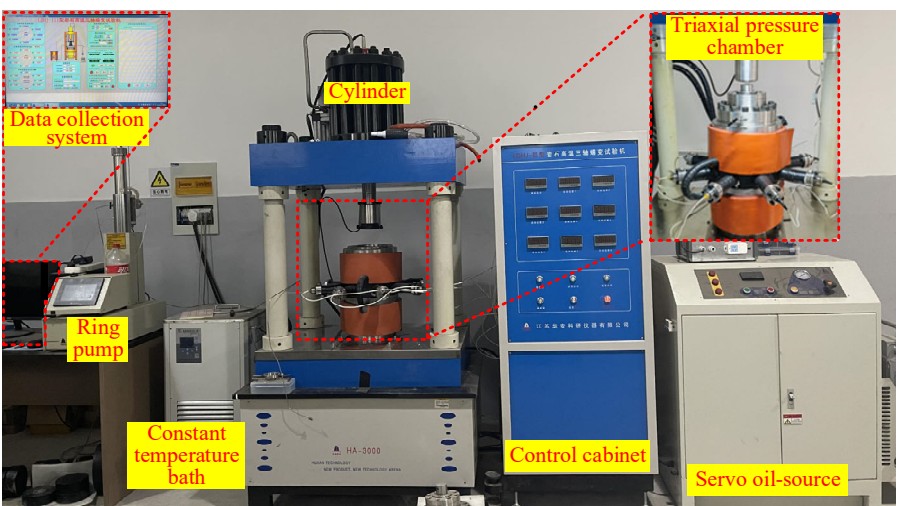

**Figure 2.** LDHJ−III high−temperature triaxial creep testing machine for rock.

## 3. Conventional Mechanical Experiment of Anthracite

### 3.1. Triaxial Compression Test

Four groups of conventional triaxial compression tests with confining pressures of 0.5, 1.0, 1.5, and 2.0 MPa were performed before the creep test. Three coal samples were tested in each group. Finally, the experimental data of a successful sample was selected from each group of experiments for analysis.

Before the start of the experiment, a layer of heat−shrinkable sleeve was wrapped on the surface of the coal sample and heated slowly and evenly with a hot air gun to tightly seal the heat−shrinkable sleeve. The coal sample was installed on the base of the triaxial pressure chamber such that the upper and lower pressure heads were consistent with the coal sample in the vertical direction. Subsequently, water was injected into the triaxial chamber to drain the air in the chamber. The confining pressure was applied to the preset stress level and maintained at a constant value. Then, the axial pressure was applied at a rate of 50 N/s until the coal sample was unstable and destroyed. After saving the experimental data, the pressure was relieved, the water was drained, and the control panel was cleared for the next group of experiments.

### 3.2. Conventional Mechanical Parameters of Anthracite under Different Confining Pressures

Figure 3 shows the complete stress–strain curves of anthracite during the triaxial compression tests with different confining pressures. The triaxial compressive strength of anthracite increased with an increase in the confining pressure. When the confining pressures were 0.5, 1.0, 1.5, and 2.0 MPa, the corresponding compressive strengths were 17.31, 21.88, 27.10, and 31.50 MPa, respectively. In the initial stage, under the action of triaxial stress, the primary microcracks and micropores in the coal samples were compacted and closed, and nonlinear strain occurred. The coal samples entered the elastic strain stage with an increase in the axial stress. The stress threshold in this stage accounted for approximately 20–40% of the triaxial compressive strength. With the continuous crack propagation and damage accumulation, the coal samples began to undergo plastic deformation and entered the strain−hardening stage. When the axial stress exceeded the peak strength, the elastic energy of the coal samples was released in large quantities, the axial stress decreased and the axial strain increased rapidly. Moreover, strain softening occurred, resulting in instability and failure of the sample.

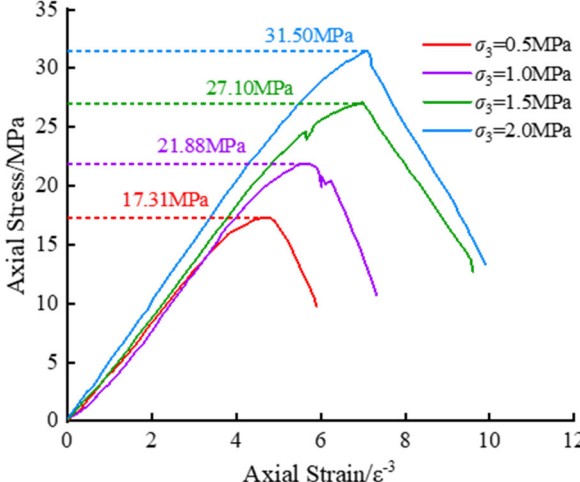

**Figure 3.** Complete stress–strain curves of anthracite during the triaxial compression tests under different confining pressures.

Elastic modulus and peak strain are important parameters reflecting the deformation performance of coal and rock mass. In rock mechanics, elastic modulus is defined as the ratio of stress and strain of coal or rock mass in the elastic deformation stage [25]. Therefore, the elastic modulus of anthracite under different confining pressures was calculated by

intercepting the stress and strain in the peak stress front deformation stage from the triaxial complete stress–strain curve. The stress and strain corresponding to the peak point of the total stress−strain curve of anthracite (Figure 3) are the instantaneous compressive strength ($\sigma_p$) and peak strain ($\varepsilon_{1c}$), as shown in Table 1. In Table 1, $\sigma_3$ represents the confining pressure and $\sigma_c$ refers to the peak strength of the deviatoric stress, which is the difference between $\sigma_p$ and $\sigma_3$.

**Table 1.** Mechanical parameters of the anthracite samples under different confining pressures.

| Sample | $\sigma_3$/MPa | $\sigma_p$/MPa | $\sigma_c$/MPa | $E_s$/GPa | $\varepsilon_{1c}/10^{-3}$ |
|---|---|---|---|---|---|
| WY−S01# | 0.5 | 17.31 | 16.81 | 4.23 | 4.78 |
| WY−S02# | 1.0 | 21.88 | 20.88 | 4.26 | 5.73 |
| WY−S03# | 1.5 | 27.10 | 25.60 | 4.65 | 6.99 |
| WY−S04# | 2.0 | 31.51 | 29.51 | 5.18 | 7.09 |

Figure 4 shows the exponential function fitting relationship between the instantaneous compressive strength, elastic modulus, and peak strain of anthracite under different confining pressures. The figure shows that in the conventional triaxial compression experiment, the compressive strength of anthracite increased linearly with confining pressure and that the elastic modulus and peak strain of anthracite increased with an increase in the confining pressure, showing an exponential function increasing trend. As the confining pressure increased, the elastic modulus of anthracite increased faster and the peak strain increased slower.

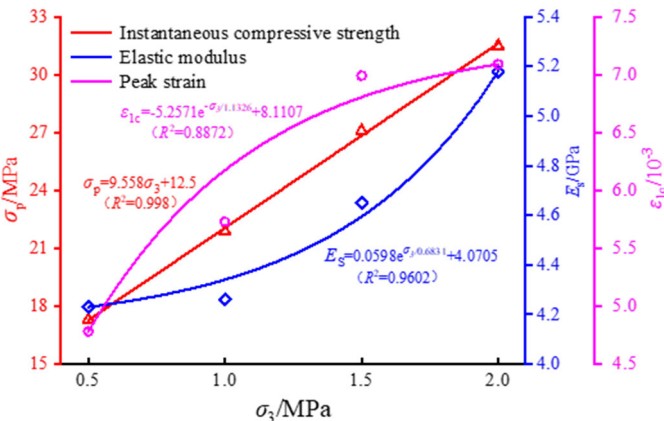

**Figure 4.** Fitting curves of instantaneous compressive strength, elastic modulus, and peak strain of anthracite under different confining pressures.

## 4. Creep Test of Anthracite

### 4.1. Test Program

The triaxial creep test of anthracite was divided into four groups according to the confining pressure grade, with two coal samples in each group. The triaxial creep test of eight samples was performed with the staged loading mode. Figure 5 shows the loading path. During the experiment, confining pressure was first applied to the specimen to a predetermined value. Subsequently, axial pressure, with the same amount as that of the confining pressure, was applied to the specimen such that the specimen was temporarily in hydrostatic pressure. After setting the other test parameters, axial stress was applied to the specimen in a stepwise manner until the specimen was unstable and destroyed. Each load was retained for 2 d.

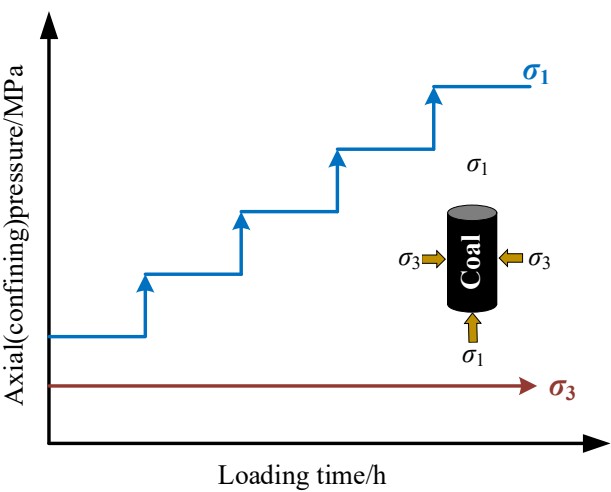

**Figure 5.** Schematic of the load application scheme for the triaxial creep test of anthracite.

*4.2. Creep Test Results*

According to the test results, the ideal test data were extracted from each group of experiments, and the "axial strain–time" curves of the coal samples under different confining pressures were obtained, as shown in Figure 6.

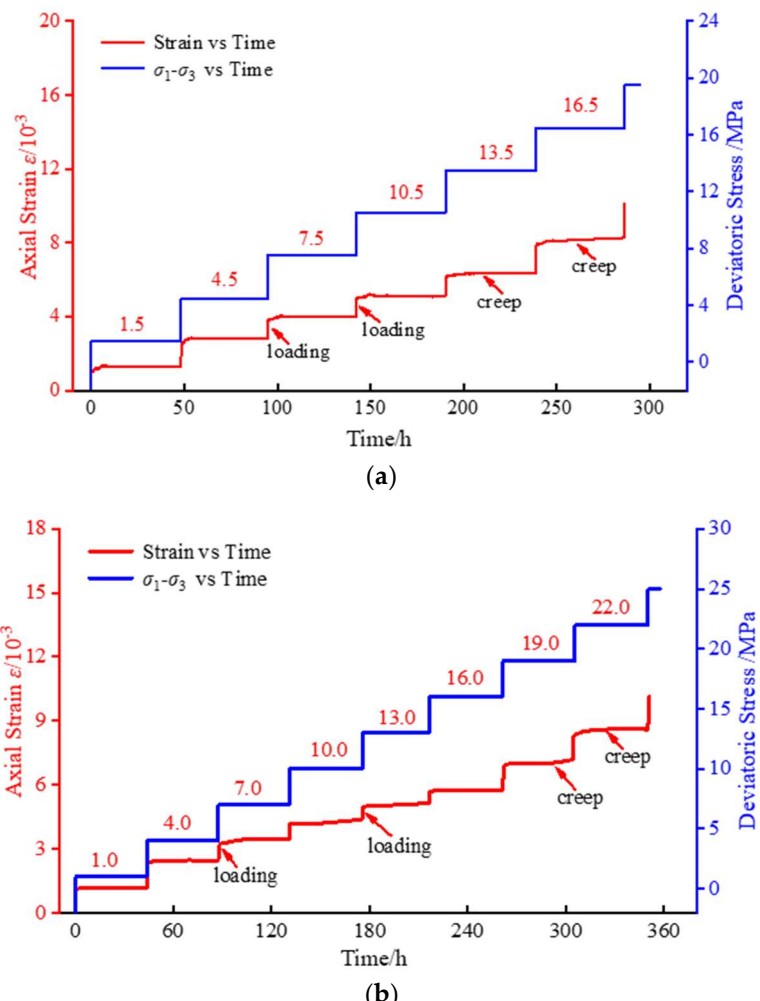

**Figure 6.** *Cont.*

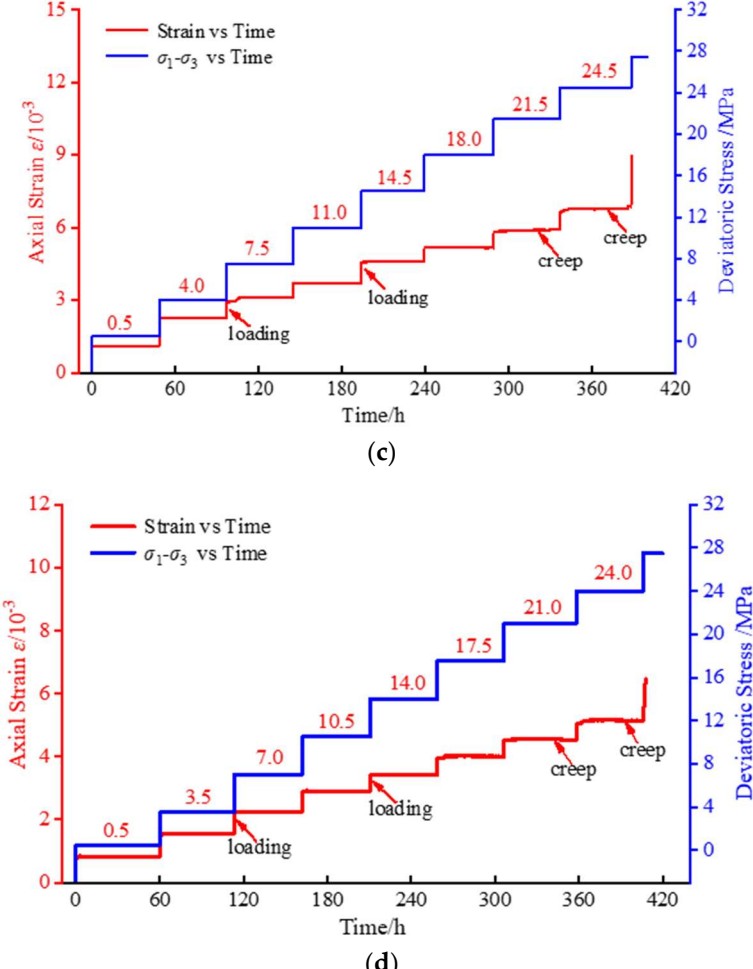

**Figure 6.** Anthracite creep test curves under different confining pressures and deviatoric stresses. (**a**) $\sigma_3$ = 0.5 MPa; (**b**) $\sigma_3$ = 1.0 MPa; (**c**) $\sigma_3$ = 1.5 MPa; (**d**) $\sigma_3$ = 2.0 MPa.

Figure 6 shows that under different confining pressures, the anthracite samples produced a particular amount of instantaneous elastic strain at the moment of loading; then, creep deformation gradually occurred with an increase in time under the action of axial stress. When the axial stress level was low, the creep rate decreased with an increase in time and gradually tended to zero, the deformation of coal samples no longer increased, and anthracite underwent only instantaneous elastic strain and decay creep. With an increase in the axial stress level, the creep rate gradually decreased with an increase in time after an instantaneous elastic deformation and finally tended to a non−zero value and remained unchanged. At this time, the anthracite entered the steady creep stage and had the characteristics of instantaneous elastic strain, attenuation creep, and steady creep. When the stress level was higher (exceeding the long−term strength of anthracite), the creep rate of anthracite was higher, and the accelerated creep occurred in a short time until the coal sample was destroyed.

Figure 6 also shows that the instantaneous elastic deformation of anthracite accounted for a large proportion of creep deformation. The axial creep deformation of anthracite was small when the stress level was low. Moreover, the axial creep increased slowly with an increase in the stress level. Taking the deformation curve of anthracite at a 1.5 MPa confining pressure as an example, under the first four stress levels, the coal samples underwent only instantaneous elastic deformation and attenuation creep stage, and there was no steady creep deformation. Under the last four axial stress levels, the coal samples underwent attenuation creep and then entered the steady creep deformation stage. With an increase in the stress levels, the steady creep rates increased continuously, with values of

$0.022 \times 10^{-5}$/h, $0.121 \times 10^{-5}$/h, $0.244 \times 10^{-5}$/h, and $0.691 \times 10^{-5}$/h. When the last load was applied to the specimen, the strain rate increased continuously after a long period of steady creep until the specimen was unstable and destroyed; that is, the specimen experienced an accelerated creep stage.

The confining pressure level directly affects the creep deformation and failure stress of anthracite. The larger the confining pressure, the stronger the lateral action of anthracite, the smaller the creep deformation, and the higher the stress level during failure. When the confining pressures were 0.5, 1.0, 1.5, and 2.0 MPa, the corresponding axial stress levels of the coal samples during failure were 17.0, 23.0, 26.0, and 26.0 MPa, respectively. The creep failure strength of anthracite decreased under different confining pressures compared with the conventional triaxial compressive strength of anthracite in Table 1; that is, the ageing effect of creep reduced the anthracite strength. Therefore, long−term strength is a key factor to be considered in studying the long−term stability of coal pillars.

### 4.3. Instantaneous Elastic Strain

Figure 7 shows the instantaneous elastic strain–deviatoric stress relationship curve of anthracite under different confining pressures. The figure shows that the instantaneous elastic strain was almost linear with the loading stress under each confining pressure grade with an increase in the deviatoric stress, and the slopes of linear fitting were 0.446, 0.322, 0.222, and 0.173 with different confining pressures. The linear fitting effect was good, and the fitting coefficients were greater than 0.99; that is, the instantaneous strain of the sample increased continuously with the axial stress. Under the same deviatoric stress level, the smaller the confining pressure, the greater the instantaneous elastic strain. This implies that the confining pressure significantly influenced the lateral constraint of the coal samples. In addition, the smaller the confining pressure, the smaller the axial deformation. Moreover, the ability of the anthracite specimens to resist instantaneous elastic deformation increased with an increase in the confining pressure level. Figure 7 also shows that the smaller the confining pressure, the greater the growth rate of the instantaneous elastic deformation with the axial stress level. The change curve of the instantaneous elastic strain growth rate with confining pressure was plotted according to the slope of the fitting line, as shown in Figure 8. The figure shows that the growth rate of the instantaneous elastic strain of anthracite decreased gradually with an increase in the confining pressure, and the relationship between them decreased exponentially.

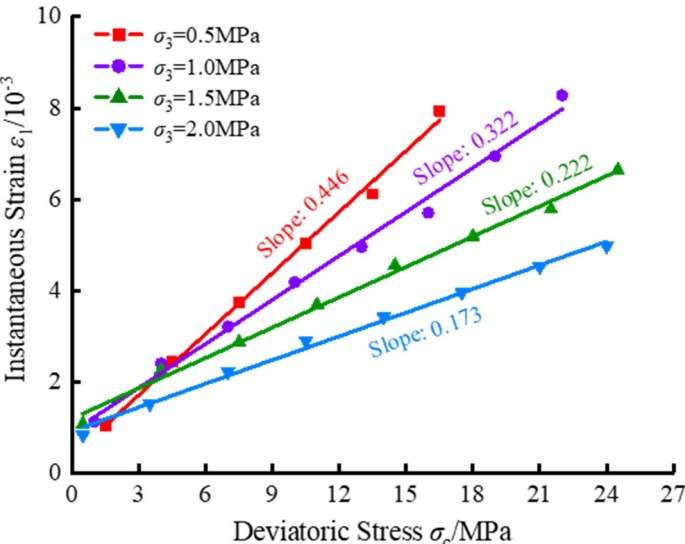

**Figure 7.** Relationship curve between the instantaneous elastic strain and deviatoric stress under different confining pressures.

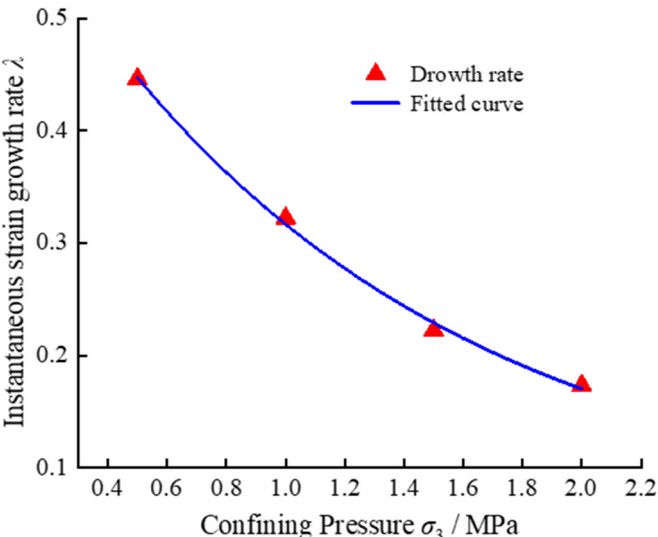

**Figure 8.** Variation curve of the instantaneous elastic strain growth rate of anthracite with confining pressure.

### 4.4. Creep Strain Rate

Figure 9a shows the three−stage creep characteristics of the anthracite sample under the loading condition with a 0.5 MPa confining pressure and 16.5 MPa single−order deviator stress. Because the deviatoric stress level was very close to the peak strength of the coal sample, the accelerated creep occurred instantaneously when the coal sample was loaded at the next stage after creep for a particular time, leading to its instability and failure. Figure 9a also shows that the instantaneous elastic deformation occurred at the moment of loading and then entered the creep stage. The entire creep process lasted approximately 48 h. The initial creep deformation was apparent. The deformation rate decelerated and decreased. The axial strain rate decreased from $0.5 \times 10^{-3}$ min$^{-1}$ to $0.1 \times 10^{-4}$ min$^{-1}$ and then stabilised. Stable creep occurred approximately 11 h after loading, showing apparent linear creep characteristics. Creep deformation increased linearly. The axial creep rate was maintained at approximately $0.1 \times 10^{-4}$ min$^{-1}$. Stable creep lasted approximately 37 h. The deformation rate increased sharply when a greater load was applied, and the coal sample became unstable and was destroyed.

Figure 9b shows the creep deformation and rate characteristic curves of the anthracite sample under a 1.5 MPa confining pressure and 24.5 MPa last−order deviatoric stress. The figure shows that the specimen exhibited three apparent stages of creep characteristics: attenuation creep, stable creep, and accelerated creep. The total creep duration was approximately 52 h. The amount of creep deformation was large during the decay creep period; however, the deformation rate decreased rapidly. After approximately 8 h, it entered the stable creep stage. The strain curve increased linearly and slowly in the steady creep stage, and the creep rate was relatively maintained at approximately $0.5 \times 10^{-5}$ min$^{-1}$. After the steady creep stage lasted approximately 39.2 h, the strain speed increased slowly at first and then increased sharply. The creep deformation also increased slowly at first and then increased rapidly until the coal sample was destroyed; that is, the sample experienced accelerated creep.

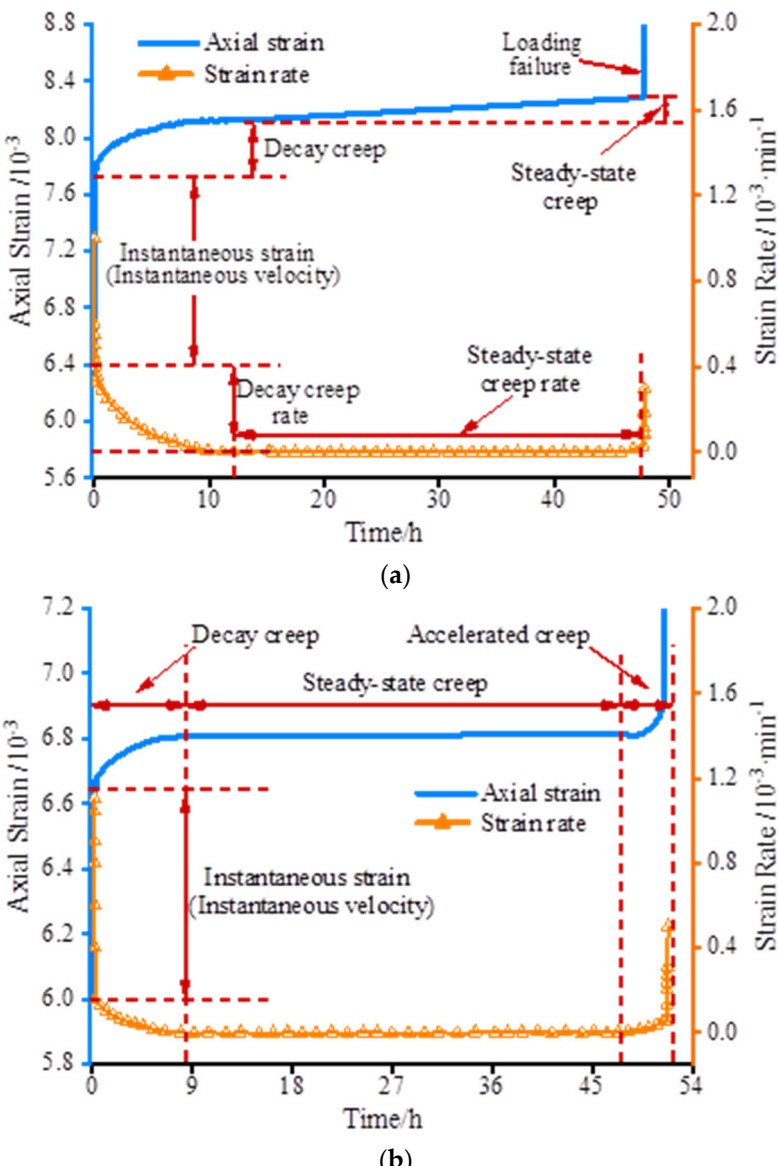

**Figure 9.** Creep rate variation of the anthracite samples with time (taking the last level as an example). (**a**) W−3 coal sample, $\sigma_1$ = 26 MPa, $\sigma_2$ = $\sigma_3$ = 1.5 MPa; (**b**) W−4 coal sample, $\sigma_1$ = 26 MPa, $\sigma_2$ = $\sigma_3$ = 2.0 MPa.

## 5. Long–Term Strength of Anthracite

The long−term strength of coal is an important evaluation index of the stability and safety of coal pillar length in coal mines. The long−term strength of coal is the critical stress value when the steady creep changes to the unsteady creep during creep deformation. When the stress level is lower than the critical value, coal remains intact regardless of the stress loading duration. By contrast, creep failure of coal eventually occurs when the stress level is higher than the critical value. The commonly used methods for determining the long−term strength of rock include the transition creep, isochronous stress–strain curve cluster, and steady creep rate methods.

### 5.1. Transition Creep Method

According to the transitional creep (TC) method, rocks have a stress threshold. Rocks will not be damaged when the external load is lower than this value; that is, the instantaneous strain and attenuation creep stage will only appear in the creep loading process. By contrast, when the external load is higher than this stress threshold, the internal damage

in rocks will accumulate continuously with the creep loading time until instability failure occurs; that is, the rocks undergo a stable or accelerated creep stage. The long−term strength of rock is defined as the maximum load that rock can bear without stable creep. The long−term strength of rock is determined by observing the change in the creep rate, which is the governing concept of the TC method.

Figure 10 shows the axial creep curve of axial compression under staged loading under different confining pressures. Because there were many loading grades when the confining pressure was 2.0 MPa, the TC method was used to determine and analyse its long−term strength. The slopes of the fitting curves were 0, 0, and $2.439 \times 10^{-5}$ (with corresponding axial stress levels of 19.5 MPa, 23 MPa, and 26 MPa, respectively), obtained by fitting the data of the first three stress levels before failure of the anthracite samples. Steady creep occurred only under the last stress; therefore, 26 MPa can be considered as the long−term strength of anthracite under this confining pressure. Similarly, the long−term strength of anthracite at confining pressures of 0.5, 1.0. and 1.5 MPa were 14 MPa, 20 MPa, and 23 MPa, respectively.

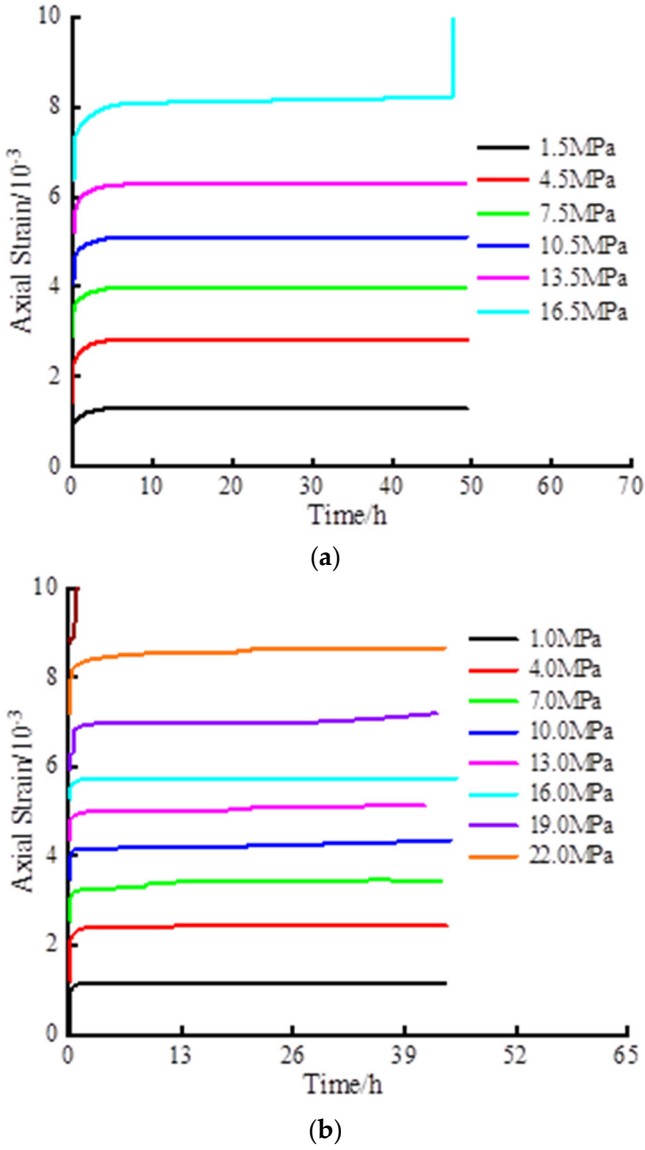

**Figure 10.** *Cont*.

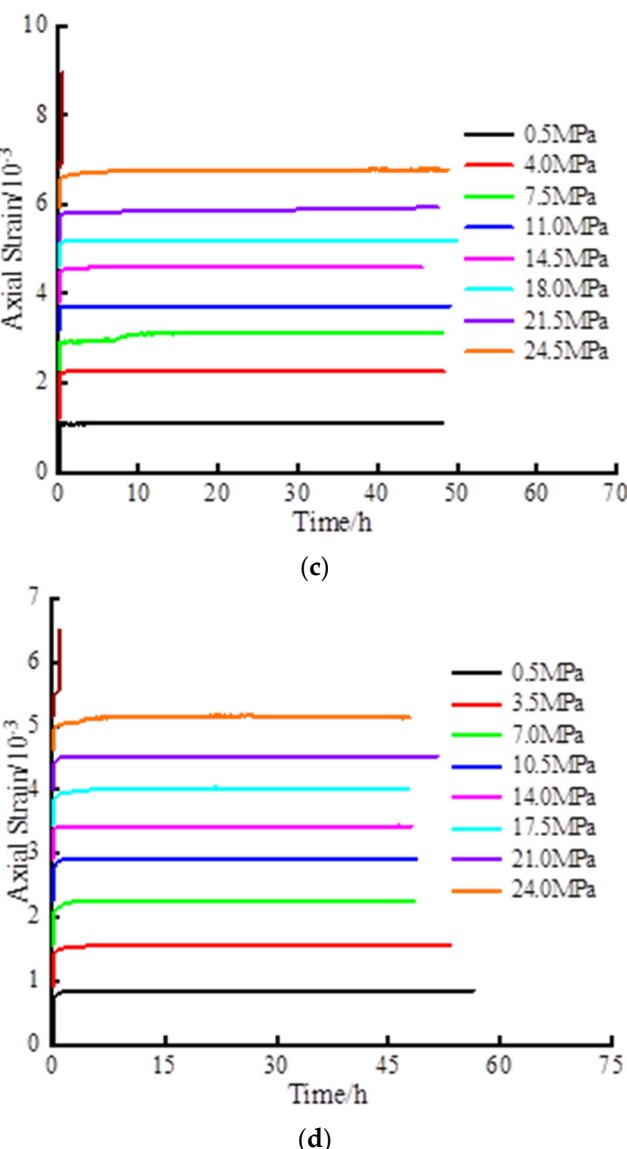

**Figure 10.** Axial creep curves of graded loading under different confining pressures. (**a**) $\sigma_3$ = 0.5 MPa; (**b**) $\sigma_3$ = 1.0 MPa; (**c**) $\sigma_3$ = 1.5 MPa; (**d**) $\sigma_3$ = 2.0 MPa.

*5.2. Isochronous Curve Method*

The isochronous curve (IC) method involves drawing the relationship curve between the creep strain and stress corresponding to equal times under different stress levels according to the strain–time curve obtained by graded loading and determining the long−term strength of rock using the stress value corresponding to the apparent inflection point of the isochronous curve. The IC method is a widely used method for determining the long−term strength of rock. Figure 11 shows the stress–strain isochronous curves of anthracite under different confining pressures obtained through the IC method.

The figure shows that the isochronous curves changed from linear to nonlinear with an increase in the axial stress level and that the degree of nonlinear bending increased continuously. Moreover, the curves gradually deviated from the strain axis, and the isochronous curves changed from dense to sparse. The inflection point was identified according to the principle of the IC method. The stress corresponding to this point was the long−term strength of anthracite. The long−term strengths of anthracite under different confining pressures were 12.91 MPa, 16.43 MPa, 18.04 MPa, and 21.75 MPa.

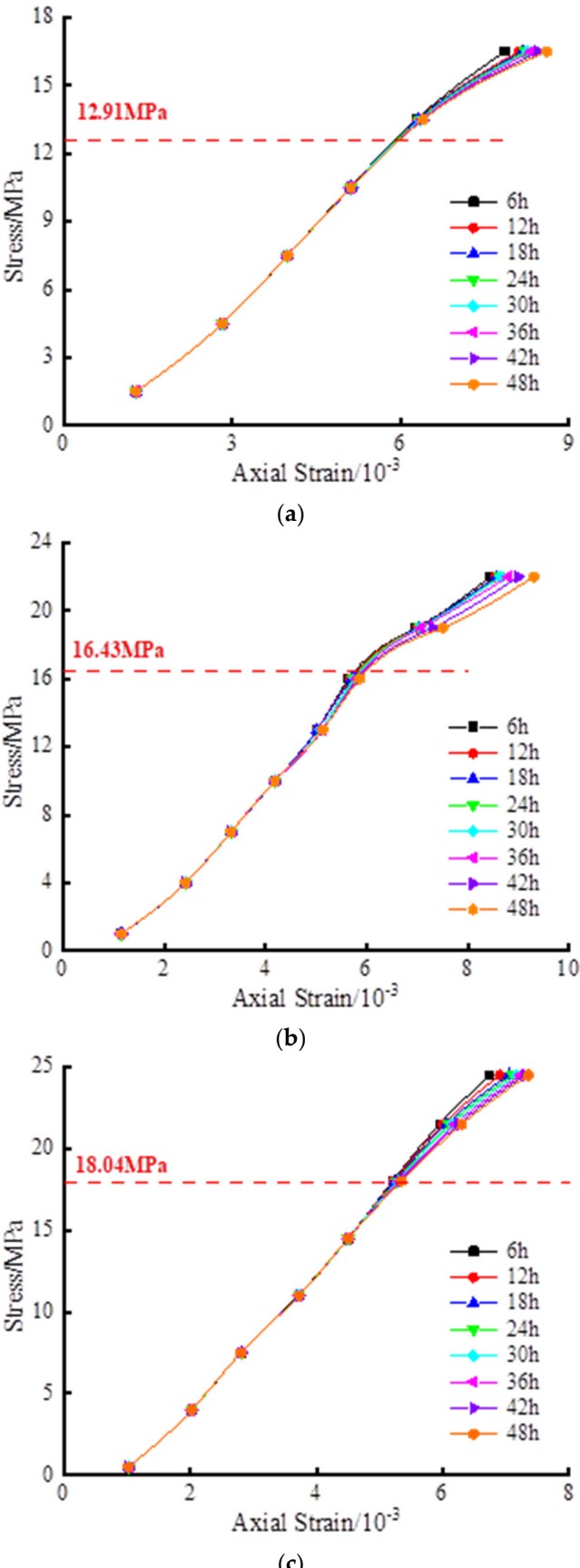

**Figure 11.** *Cont.*

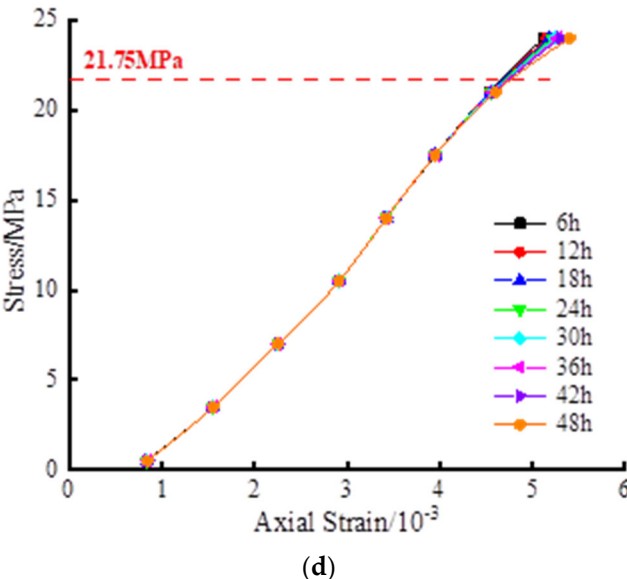

(**d**)

**Figure 11.** Stress–strain isochronous curve clusters under different confining pressures. (**a**) $\sigma_3$ = 0.5 MPa; (**b**) $\sigma_3$ = 1.0 MPa; (**c**) $\sigma_3$ = 1.5 MPa; (**d**) $\sigma_3$ = 2.0 MPa.

*5.3. Improved Steady−State Creep Rate Method*

The principle of the steady−state creep rate method is relatively the same as that of the transitional creep method. That is, when the axial load was lower than the long−term strength, the specimen only underwent attenuation creep, and the creep rate remained unchanged at zero with the progress of time, keeping the rock intact. In contrast, when the applied load was higher than the long−term strength, the sample entered the accelerated creep stage after a period of steady creep and lost stability. The steady creep rate changed from zero to non−zero and increased gradually with the axial load, whereas the steady creep time decreased gradually. Therefore, the long−term strength of rock is defined as the maximum load with zero steady creep rate. However, in the creep test, owing to the large differences between the loading stress levels, determining the minimum stress level with a zero steady creep rate was subjective and resulted in a large error.

The uniaxial creep curve of anthracite in Section 4.2 shows that when the stress level is low, the anthracite only undergoes instantaneous elastic deformation and attenuation creep, and long−term stability is maintained. In contrast, when the stress level is high, instantaneous elastic deformation, attenuation creep, and steady creep occur; the deformation increases with time; and creep failure occurs. Thus, the coal sample will lose stability and fail after creeping for a particular time when the non−zero steady−state creep rate increases continuously. Therefore, the long−term strength of coal is the critical stress of the steady creep rate of the coal sample. This study improved the steady−state creep rate method, and this new method is referred to as the improved steady−state creep rate method (ISCR). The method involves fitting the steady−state creep rate of anthracite under high stress levels, establishing the functional relationship between the axial stress and steady−state creep rate, and obtaining the long−term strength of anthracite. Figure 12 shows the steady−state creep rate fitting curve of anthracite in a high−stress range under different confining pressures.

Figure 12 shows that the steady creep rate of anthracite was zero when the axial stress was low. In contrast, the steady creep rate of anthracite increased exponentially with the stress when the stress level was high. The stress $\sigma$ corresponding to the fitting function $\varepsilon_\infty \to 0$ is the long−term strength of anthracite under the confining pressure. The long−term strengths of anthracite under different confining pressures calculated through the ISCR method were 13.46 MPa, 17.01 MPa, 20.85 MPa, and 23.97 MPa.

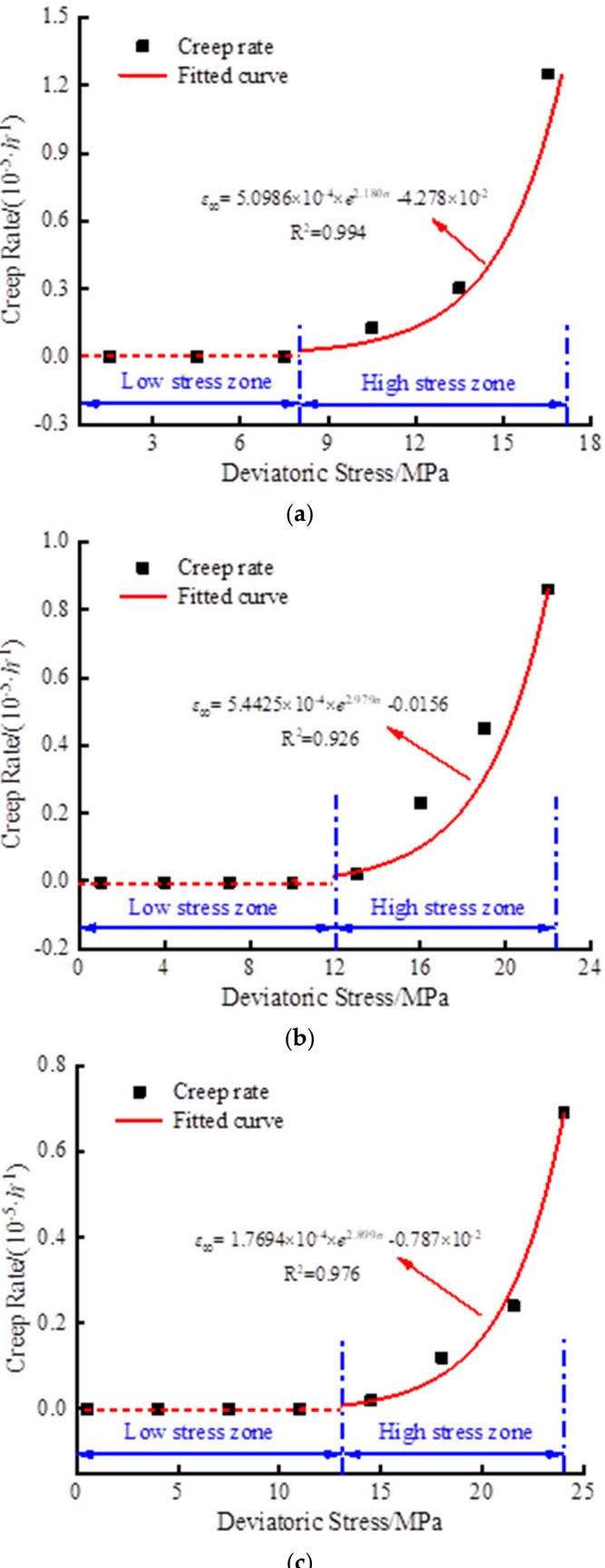

**Figure 12.** *Cont.*

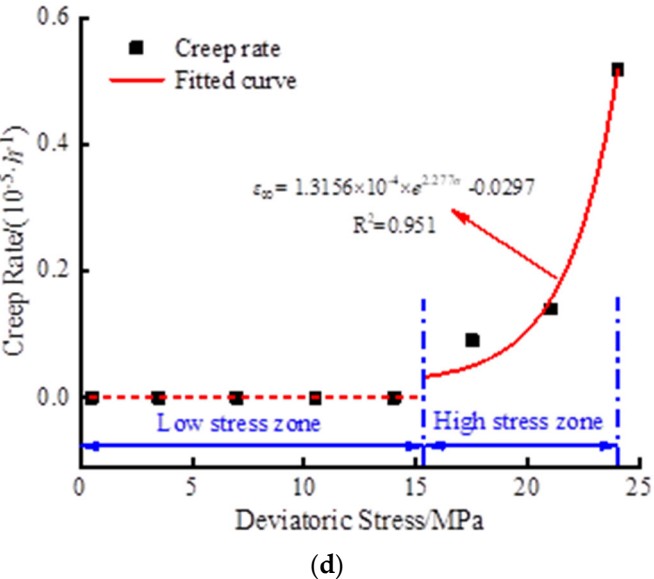

**(d)**

**Figure 12.** High−stress axial steady creep rate of anthracite under different confining pressures. (**a**) $\sigma_3$ = 0.5 MPa; (**b**) $\sigma_3$ = 1.0 MPa; (**c**) $\sigma_3$ = 1.5 MPa; (**d**) $\sigma_3$ = 2.0 MPa.

*5.4. Comparative Analysis*

Based on the creep test results of anthracite under different confining pressures, the long−term strength of anthracite was obtained through the abovementioned methods. The TC method is simple in terms of data processing, and only the axial strain–time curves under various stresses need to be observed or fitted to determine the critical stress from stable to unstable creep. This method is influenced by the stress level of the creep test. The accurate long−term strength of anthracite could not be determined; however, an approximate value or approximate interval of long−term strength can be determined. In addition, the stable creep stage could not be observed intuitively in the triaxial creep of anthracite in this test; therefore, the long−term strength value of anthracite determined through this method needs further evaluation. The IC method is complicated in terms of data curve processing. First, the entire strain–time curve of anthracite must be transformed into the strain–time curve under various stresses. Subsequently, an appropriate period is selected, and the curve is further transformed into stress–strain isochronous curves. In this study, when this method was used to determine the long−term strength of anthracite, the inflection point of the curve cluster was not apparent, and the obtained long−term strength of anthracite was significantly affected by subjective factors. The ISCR method maximises the characteristic that anthracite goes through a steady−state creep stage with an increasing creep rate when the axial stress level is high. The steady−state creep rate–stress curve in high stress level intervals was plotted, the long−term strength of anthracite was determined according to the functional relationship of the fitting curve, and the error caused by the subjective judgement of determining the sudden change point of creep rate was eliminated. This method can determine the long−term strength of anthracite reasonably and accurately. Table 2 shows the long−term strength of anthracite under different confining pressures determined by each method.

Figure 13 compares the long−term strength and instantaneous compressive strength of anthracite determined through the abovementioned methods and shows the variation law with the increase in the confining pressure grade. The figure shows that similar to the instantaneous strength, the long−term strength of anthracite determined through the above methods increased with an increase in the confining pressure, which was less than its instantaneous strength. The long−term strength to instantaneous strength ratios of anthracite under each confining pressure grade were 0.75–0.81%, 0.75–0.91, 0.70–0.85, and 0.71–0.79. The comparison of the three methods shows that the TC method obtained the highest long−term strength, followed by the ISCR method and finally the IC method.

**Table 2.** Long−term strength of anthracite under different confining pressures determined using each method.

| $\sigma_3$/MPa | Transition Creep Method | Isochronous Curve Method | Improved Steady−State Creep Rate Method |
|---|---|---|---|
| 0.5 | 14 MPa | 12.91 MPa | 13.46 MPa |
| 1.0 | 20 MPa | 16.43 MPa | 17.01 MPa |
| 1.5 | 23 MPa | 18.04 MPa | 20.85 MPa |
| 2.0 | 26 MPa | 21.75 MPa | 23.97 MPa |

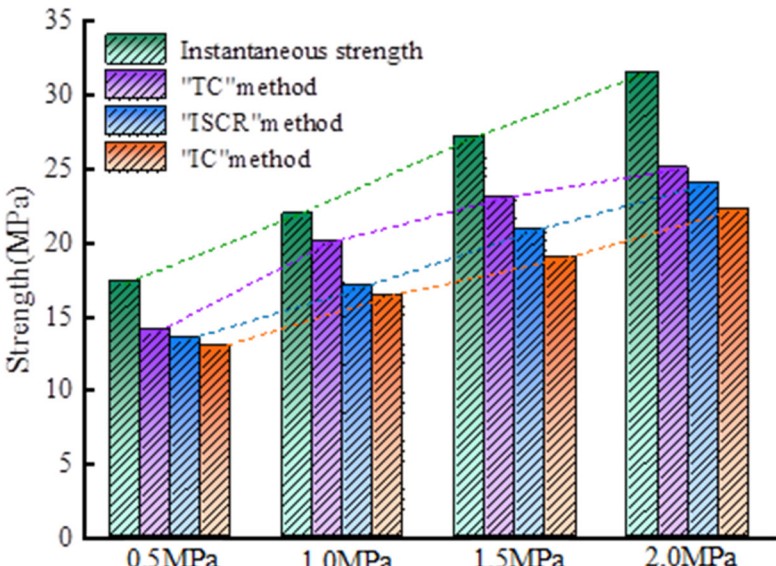

**Figure 13.** Comparison between long−term strength and instantaneous strength of anthracite determined through different methods and its variation with confining pressure.

The long−term strength value determined through the ISCR method is considered the optimal strength value of anthracite. This study introduces the concept of the ageing strength drop ($\Delta\sigma_\infty$) of anthracite; that is, the difference between the instantaneous strength and long−term strength of anthracite ($\Delta\sigma_\infty = \sigma_p - \sigma_\infty$). Figure 14 shows the variation of the long−term strength drop of anthracite with confining pressure.

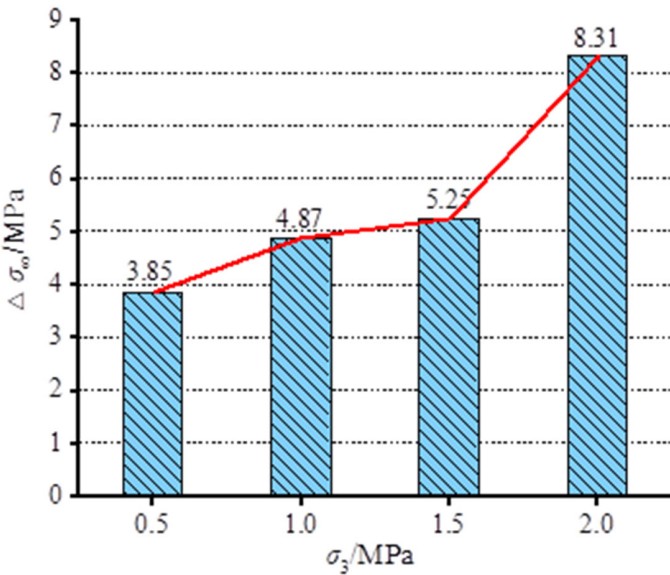

**Figure 14.** Ageing strength drop of anthracite under different confining pressures.

## 6. Conclusions

This study performed a triaxial compression creep test to examine the creep behaviour of anthracite samples under different confining pressures. In addition, different methods were used to study the long−term strength of anthracite under different confining pressures. The advantages and disadvantages of each method were compared and analysed to comprehensively determine the long−term strength of anthracite samples. The primary conclusions of the study are summarised as follows:

(1) The conventional triaxial compression experiments showed that the elastic modulus and peak strain of anthracite increased with an increase in the confining pressure, exhibiting an exponential function increasing trend. The fitting degrees with confining pressure were 0.9602 and 0.8872.

(2) The creep characteristics of anthracite were closely related to the stress level. Under low stress levels, the anthracite samples exhibited only instantaneous elastic strain and attenuation creep characteristics. In contrast, the anthracite samples exhibited instantaneous elastic deformation, attenuation creep, and steady creep under high stress levels. When the stress reaches a particular level, accelerated creep occurred until failure.

(3) The instantaneous elastic strain and loading stress of anthracite under different confining pressures increased almost linearly, and the linear fitting coefficients were all greater than 0.99. The larger the confining pressure, the greater the lateral constraint on anthracite and the smaller the axial strain. The growth rate of the instantaneous elastic strain of anthracite decreased exponentially with an increase in the confining pressure.

(4) The long−term strength of anthracite determined through each method increased with an increase in the confining pressure, which was less than its instantaneous strength. The long-term strength to instantaneous strength ratio under each confining pressure class ranged from 70% to 91%. The TC method obtained the highest long-term strength, followed by the ISCR method and the IC method.

(5) The functional relationship between the axial stress and steady-state creep rate was established by fitting the steady−state creep rate of anthracite under high stress levels. The threshold value of the steady−state creep rate in the high−stress−level area was proposed as the optimal long-term strength of anthracite. The improvement of this method provides a relevant reference for research on the long-term strength of coal and rock mass.

(6) The ageing characteristic of creep reduced the strength of anthracite. The long−term strength determined through the ISCR method was considered as the optimal long-term strength value of anthracite. The ageing strength drop values of anthracite under different confining pressures were 3.85 MPa, 4.87 MPa, 5.25 MPa, and 8.31 MPa.

**Author Contributions:** Conceptualization, J.Y., X.Z. and S.Y.; methodology, X.Z.; validation, X.Z., X.S. and K.W.; formal analysis, J.Y.; investigation, J.H.; data curation, J.Y., S.Y. and J.H.; writing—original draft preparation, J.Y.; writing—review and editing, J.Y. and X.Z.; funding acquisition, X.Z. and K.W. All authors have read and agreed to the published version of the manuscript.

**Funding:** This work was financially supported by the National Natural Science Foundation of China (No. 51704204, No. 51974194, No. 51904197, No. 52104097) and the basic research program of Shanxi province, China (No. 20210302123147, No. 20210302124352).

**Data Availability Statement:** Not applicable.

**Acknowledgments:** This work was financially supported by the National Natural Science Foundation of China (No. 51704204, No. 51974194, No. 51904197, No. 52104097) and the basic research program of Shanxi province, China (No. 20210302123147, No. 20210302124352).

**Conflicts of Interest:** The authors declare no conflict of interest.

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
