# Peer review of "Experimental Study on Creep Characteristics and Long–Term Strength of Anthracite"

_processes, doi:10.3390/pr11030947_

Round 1
Reviewer 1 Report
This paper is an experimental study on anthracite's strength and long-term creep deformation, and the following needs to be modified.
M) It is recommended to calculate the linear estimation formula of the elastic modulus and strength to confining pressure so that the experimental results in Figure 3 apply to the classical plasticity model (Figure 4 and Table 1).
M) In Fig. 7, only the slope of the deviatoric stress-instantaneous strain to the confining pressure is needed, and additional formulas are considered meaningless. And it is desirable to simplify the estimation equation in Figure 8.
M) Rather than experimental results for each confining pressure in Figure 12 and Table 2, it is preferable to normalize the x-axis (deviatoric stress) with confining pressure and integrate all experimental results.
Author Response
Dear Editors and reviewers,
Thank you for your letter and the comments concerning our manuscript entitled “Experimental study on creep characteristics and long-term strength of anthracite” (Manuscript ID: processes-2286292). All your comments are valuable and helpful for improving our paper. We have considered the reviewers’ comments and recommendations carefully and have made detailed revisions to our previous manuscript. The responses to the reviewers’ comments are listed below.
Response to the comments of reviewer 1:
- It is recommended to calculate the linear estimation formula of the elastic modulus and strength to confining pressure so that the experimental results in Figure 3 apply to the classical plasticity model (Figure 4 and Table 1).
Authors’ response: Thank you very much for the valuable comments. The comments are very useful for improving our manuscript. We would like to express our great appreciation to you. According to your constructive comments and suggestions, we have supplemented the curve of peak strength of anthracite with confining pressure, and fitted the curve, as shown in Fig. 4. In our revision, we supplemented relative information, as can be seen in Page 5, lines 172-173.
- In Fig. 7, only the slope of the deviatoric stress-instantaneous strain to the confining pressure is needed, and additional formulas are considered meaningless. And it is desirable to simplify the estimation equation in Figure 8.
Authors’ response: Thanks a lot for this kind comments. According to your comments, we have deleted the additional formulas in Fig. 7, and simplified the estimation equation in Fig. 8 at the same time, as can be seen in Figures 7 and 8. In our revision, we supplemented relative information, as can be seen in Page 8, lines 243-244.
- Rather than experimental results for each confining pressure in Figure 12 and Table 2, it is preferable to normalize the x-axis (deviatoric stress) with confining pressure and integrate all experimental results.
Authors’ response: Thanks for your kind comments. According to your comments, we have integrated all experimental results of the long-term strength of anthracite under different confining pressures deter-mined by each method, as can been seen in Table 2. Table 2 in Section 5.3 was modified and then moved to Section 5.4. And we supplemented relative information in Page 16, lines 424-425.
Thanks again for your valuable comments and advice, your deep comments and practical advice contributed a lot to improve the quality of this article. We are looking forward to hearing from you regarding our revised manuscript. We would be glad to respond to any further questions and comments that you may have.
Best regards.
Sincerely yours,
Corresponding author: Xiaoqiang Zhang
Taiyuan University of Technology
E-mail: tyzxq2009@163.com
Reviewer 2 Report
This is a good paper on the creep and strength properties of anthracite. I have a couple of questions.
1. How did you decide the range of confining pressure from 0.5-2.0 MPa? Actual confining pressure within a coal pillar can be higher than 2.0 MPa.
2. How would the creep property and long-term coal strength be affected by the size of coal specimens?
A few editorial comments are shown in the text.

Author Response
Dear Editors and reviewers,
Thank you for your letter and the comments concerning our manuscript entitled “Experimental study on creep characteristics and long-term strength of anthracite” (Manuscript ID: processes-2286292). All your comments are valuable and helpful for improving our paper. We have considered the reviewers’ comments and recommendations carefully and have made detailed revisions to our previous manuscript. The responses to the reviewers’ comments are listed below.
Response to the comments of reviewer 2:
- How did you decide the range of confining pressure from 0.5-2.0 MPa? Actual confining pressure within a coal pillar can be higher than 2.0 MPa.
Authors’ response: Thank you very much for the valuable comments. We intended to say that the stress state when the coal pillar exists alone. The stress environment in the elastic core area of the coal pillar is generally a low confining pressure triaxial stress state. Therefore, when the triaxial creep test of anthracite was performed in this paper, the confining pressure stress level was low. In our revision, we have made relevant supplementary explanations in the introduction, and the revision is listed in Page 1-2, lines 43-46.
- How would the creep property and long-term coal strength be affected by the size of coal specimens?
Authors’ response: Thanks for your comments. In this paper, when studying the triaxial creep characteristics and long-term strength of anthracite, we conducted triaxial creep tests by processing coal into standard cylindrical samples with a diameter of Φ50mm×100mm according to the test regulations of the International Society of Rock Mechanics (ISRM).
In addition, in view of some other editorial comments that you mentioned in the text, we have revised them one by one, and the revision is listed in lines 9, 29, 108, 125, and 423.
Thanks again for your valuable comments and advice, your deep comments and practical advice contributed a lot to improve the quality of this article. We are looking forward to hearing from you regarding our revised manuscript. We would be glad to respond to any further questions and comments that you may have.
Best regards.
Sincerely yours,
Corresponding author: Xiaoqiang Zhang
Taiyuan University of Technology
E-mail: tyzxq2009@163.com